# Cerebral Folate Deficiency, Folate Receptor Alpha Autoantibodies and Leucovorin (Folinic Acid) Treatment in Autism Spectrum Disorders: A Systematic Review and Meta-Analysis

**DOI:** 10.3390/jpm11111141

**Published:** 2021-11-03

**Authors:** Daniel A. Rossignol, Richard E. Frye

**Affiliations:** 1Rossignol Medical Center, 24541 Pacific Park Drive, Suite 210, Aliso Viejo, CA 92656, USA; 2Barrow Neurological Institute at Phoenix Children’s Hospital, Phoenix, AZ 85016, USA; rfrye@phoenixchildrens.com; 3Department of Child Health, University of Arizona College of Medicine, Phoenix, AZ 85004, USA

**Keywords:** autism spectrum disorder, cerebral folate deficiency, folate receptor alpha autoantibodies, folinic acid, leucovorin

## Abstract

The cerebral folate receptor alpha (FRα) transports 5-methyltetrahydrofolate (5-MTHF) into the brain; low 5-MTHF in the brain causes cerebral folate deficiency (CFD). CFD has been associated with autism spectrum disorders (ASD) and is treated with *d,l*-leucovorin (folinic acid). One cause of CFD is an autoantibody that interferes with the function of the FRα. FRα autoantibodies (FRAAs) have been reported in ASD. A systematic review was performed to identify studies reporting FRAAs in association with ASD, or the use of *d,l*-leucovorin in the treatment of ASD. A meta-analysis examined the prevalence of FRAAs in ASD. The pooled prevalence of ASD in individuals with CFD was 44%, while the pooled prevalence of CFD in ASD was 38% (with a significant variation across studies due to heterogeneity). The etiology of CFD in ASD was attributed to FRAAs in 83% of the cases (with consistency across studies) and mitochondrial dysfunction in 43%. A significant inverse correlation was found between higher FRAA serum titers and lower 5-MTHF CSF concentrations in two studies. The prevalence of FRAA in ASD was 71% without significant variation across studies. Children with ASD were 19.03-fold more likely to be positive for a FRAA compared to typically developing children without an ASD sibling. For individuals with ASD and CFD, meta-analysis also found improvements with *d,l*-leucovorin in overall ASD symptoms (67%), irritability (58%), ataxia (88%), pyramidal signs (76%), movement disorders (47%), and epilepsy (75%). Twenty-one studies (including four placebo-controlled and three prospective, controlled) treated individuals with ASD using *d,l*-leucovorin. *d,l*-Leucovorin was found to significantly improve communication with medium-to-large effect sizes and have a positive effect on core ASD symptoms and associated behaviors (attention and stereotypy) in individual studies with large effect sizes. Significant adverse effects across studies were generally mild but the most common were aggression (9.5%), excitement or agitation (11.7%), headache (4.9%), insomnia (8.5%), and increased tantrums (6.2%). Taken together, *d,l*-leucovorin is associated with improvements in core and associated symptoms of ASD and appears safe and generally well-tolerated, with the strongest evidence coming from the blinded, placebo-controlled studies. Further studies would be helpful to confirm and expand on these findings.

## 1. Introduction

Autism spectrum disorder (ASD) is a behaviorally defined disorder that affects approximately 2% of children in the United States [1]. A number of medical comorbidities have been reported in individuals with ASD, with some studies reporting an average of 4–5 comorbidities [2], including allergic rhinitis [3], irritable bowel syndrome [3], attention deficit hyperactivity disorder [4], ophthalmological conditions [4], sleep problems [5], immune problems [5], mitochondrial dysfunction [6], gastrointestinal abnormalities [7] and epilepsy [8]. Recently, cerebral folate deficiency (CFD) has been reported in a number of studies in individuals with ASD and its treatment, *d,l*-leucovorin calcium (also known as folinic acid) has undergone investigation as a treatment for ASD [9].

The importance of folate for the development of the central nervous system (CNS) was first discovered in animal models of dietary folate deficiency where findings such as hydrocephalus [10], decreased myelin cerebroside [11] and impaired synthesis of neuronal RNA [12] were reported. In humans, studies of serum folate deficiency reported neurological findings including ataxia, nystagmus, areflexia, and confusion [13] as well as organic brain syndrome and damage to the pyramidal tracts [14]. In 1974, low levels of cerebrospinal fluid (CSF) folate were reported in patients with epilepsy that improved with tetrahydrofolate but not folic acid [15]. A study in 1976 reported 2 children with low CSF folate who developed intracranial calcifications [16]. In 1979, cerebral atrophy was reported in a 48-year-old woman with low serum and CSF folate [17]. In 1983, a case of a 23 year old woman with Kearns-Sayre syndrome and CNS deterioration was found to have decreased serum and CSF folate while taking phenytoin and improved with folic acid treatment [18].

In 2002, Ramaekers reported a novel neurometabolic syndrome in five children with low CSF 5-MTHF [19] who manifested severe neurodevelopmental symptoms including irritability, seizures, lower extremity pyramidal signs, spastic paraplegia, psychomotor retardation, dyskinesias, cerebellar ataxia, acquired microcephaly and developmental regression which occurred as young as 4 months of age. Unlike previous studies of folate deficiency associated with CNS abnormalities, this new neurometabolic disorder demonstrated below normal concentration of 5-methyltetrahydrofolate (5-MTHF), one of the active metabolites of folate, in the CSF, but normal systemic folate levels. This condition improved with folinic acid (*d,l*-leucovorin) treatment. In these patients, no genetic abnormalities were identified. Ramaekers et al., later described this condition as “idiopathic CFD” [20].

CFD occurs due to the impaired transport of folates across the blood-brain barrier. CFD is usually caused by dysfunction of the folate receptor-alpha (FRα) [21]. FRα is a receptor which has a high affinity for 5-MTHF and is found on the basolateral endothelial surface of the choroid plexus. FRα transports folates across the blood-brain barrier through adenosine triphosphate (ATP) dependent receptor-mediated endocytosis. Dysfunction of the FRα can occur through several mechanisms. In only rare cases, genetic mutations in the gene encoding for FRα (FOLR1) is a cause of CFD [22,23]. Two other mechanisms are more prevalent in causing FRα dysfunction. First, in the seminal paper describing CFD, two autoantibodies to the FRα (blocking and binding autoantibodies) were described that interfere with the function of the FRα [21]. Second, mitochondrial disease is associated with CFD since folate transportation through the FRα is dependent on ATP [24,25,26,27].

Another folate transporter, the reduced folate carrier (RFC/SLC19A1), lies on both the basolateral and apical surface of the choroid plexus and has a lower affinity for folates. *d,l*-leucovorin is a reduced (active) form of folate which can enter the CNS through the RFC and has been reported to normalize 5-MTHF levels in the CSF in individuals with CFD [21]. In some cases, clinical response to *d,l*-leucovorin is dramatic, especially if this treatment is started early in life [19,28] but sometimes improvements can be marked even in adults [29]. *d,l*-leucovorin consists of two diastereomers designated as *d*-leucovorin and *l*-leucovorin. *l*-Leucovorin (5-formyl-(6S)-tetrahydrofolate) is the biologically active isomer. It is rapidly metabolized (via 5,10-methenyltetrahydrofolate then 5,10-methylenetetrahydrofolate) to 5-methyl-(6S)-tetrahydrofolate (L-methyl-folate or 5-MTHF), which, in turn, can be metabolized via other pathways back to tetrahydrofolate and 5,10-methenyltetrahydrofolate. 5,10-methylenetetrahydrofolate is converted to 5-MTHF by an irreversible enzyme-catalyzed reduction using the cofactors FADH_2_ and NADPH. The major mechanism of transportation of folates into the brain occurs through the FRα. FRα binds to folate at lower serum concentrations than the RFC; the latter functions at relatively high concentrations of serum folate and is the predominant method of folate transport in the intestine. Genetic defects in RFC can lead to a rare disorder in intestine folate absorption and genetic abnormalities in FRα may lead to neural tube defects [19].

*d,l*-leucovorin was first approved in the United States in the 1950s and has been used continuously since then to reduce toxicities associated with folate pathway antagonists such as methotrexate (which is typically used in the treatment of osteosarcoma, other cancers and autoimmune diseases). By replenishing intracellular pools of reduced folates, *d,l*-leucovorin can counteract the toxic effects of folate pathway antagonists such as methotrexate which act by inhibiting dihydrofolate reductase (DHFR). The *d*-isomer of leucovorin is not metabolically active and is not metabolized *in vivo* to any significant degree; therefore, only the *l*-isomer can contribute to the direct replenishment of the pools of active folate cofactors. One of the main advantages reported for *d,l*-leucovorin over folic acid (pteroylmonoglutamic acid) is that folic acid is oxidized and must be reduced by DHFR to a biologically active folate in order to become active. High doses of folic acid may also block the FRα which can potentially exacerbate CFD [30]. Many individuals, including those with ASD, have polymorphisms in the DHFR gene which makes the function of this enzyme less efficient [31]. Currently, the most prescribed form of leucovorin in the United States is *d,l*-leucovorin calcium. *d,l*-Leucovorin is only form of folate that has been used to treat CFD except for one case report using 5-MTHF [32].

Since its original description, the phenotype of CFD has expanded. A number of studies have reported ASD in a subset of children with CFD [20,21,33,34,35,36]. This is not particularly surprising as two mechanisms for FRα dysfunction, FRα autoantibodies (FRAAs) and mitochondrial dysfunction, are common in ASD. Indeed, studies have reported a high prevalence of FRAAs in individuals with ASD ranging from 58% to 76% [37,38,39,40,41,42,43,44,45]. Mitochondrial dysfunction is a common medical comorbidity in ASD, with studies reporting 30–50% of individuals with ASD possessing biomarkers of mitochondrial dysfunction [6,46] and up to 80% having abnormal electron transport chain (ETC) activity in immune cells [47,48]. Treatment in some children with CFD and ASD with oral *d,l*-leucovorin has led to clinical improvements ranging from partial improvements in communication, social interaction, attention and stereotypies [21,33,35,36] to complete recovery of both neurological and ASD symptoms [21,34] in up to 21% of treated patients [44]. Of note, *d,l*-leucovorin has been reported to improve symptoms in Down syndrome [49], Rett syndrome [50,51,52] and schizophrenia [53].

This paper systematically reviews the studies examining an association between CFD and ASD, then examines the prevalence of the major cause of CFD, namely FRAAs, in ASD, typically developing (TD) siblings of individuals with ASD and their parents and TD non-related controls, followed by reviewing the evidence for treatment with *d,l*-leucovorin, the primary treatment for CFD, in individuals with ASD and any associated adverse effects (AEs).

## 2. Materials and Methods

### 2.1. Search Process

A prospective protocol for this review was developed a priori and the search terms and selection criteria were chosen to capture all pertinent publications. The search included individuals with autistic disorder, Asperger syndrome, pervasive developmental disorder-not otherwise specified (PDD-NOS) and ASD. A computer-aided search of PUBMED, Google Scholar, CINAHL, EmBase, Scopus, and ERIC databases from inception through September 2021 was conducted to identify pertinent publications using the search terms “autism”, “autistic”, “Asperger”, “pervasive”, “ASD”, and “PDD” in all combinations with “folinic acid”, “leucovorin”, “folate”, “folic”, “methyl-folate”, “5MTHF”, “levofolinic”, “folinate”, and “formyltetrahydrofolate.” The references cited in identified publications were also searched to locate additional studies.

### 2.2. Study Selection and Assessment

This systematic review and meta-analysis followed PRISMA guidelines [33], the PRISMA Checklist is found in Appendix A and the PRISMA Flowchart for *d,l*-leucovorin treatment in ASD is displayed as Figure 1. Studies were included in this systematic review if they: (a) involved individuals with ASD, and (b) either reported on the use of leucovorin in at least one individual with ASD and/or described FRAAs in at least one individual with ASD. Articles that did not present new or unique data (such as review articles or letters to the editor), and animal studies were excluded. Studies on Rett syndrome and Childhood Disintegrative Disorder were also excluded. One reviewer (DR) screened titles and abstracts of all potentially relevant studies for identification purposes. Both reviewers then examined each identified study in-depth and assessed factors such as the risk of bias. As per standardized guidelines [54], selection, performance detection, attrition, and reporting biases were considered.

As a result of the in-depth review, several studies were excluded from further consideration. One study [55] reported on a child with “autistic personal characteristics” who had CFD, but it was unclear whether this child was diagnosed with ASD. Two studies reported treatment of children with ASD using **folic acid** [56,57]. Finally, one study [43] used an unvalidated, non-clinical FRAA assay in which the functional significance of the autoantibody is unknown.

### 2.3. Meta-Analysis

MetaXL Version 5.2 (EpiGear International Pty Ltd., Sunrise Beach, QLD, Australia) was used with Microsoft Excel Version 16.0.12827.20200 (Redmond, WA, USA) to perform the meta-analysis. Random-effects models, which assume variability in effects from both sampling error and study level differences [58,59], were used to calculate pooled prevalence and odds ratios. The Luis Furuya-Kanamori (LFK) Index derived from Doi plots was reviewed for significant asymmetries (>±2) in the prevalence distribution when there were three or more studies [60,61]. Cochran’s Q was calculated to determine heterogeneity of effects across studies and, when significant, the I^2^ statistic (Heterogeneity Index) was calculated to determine the percentage of variation across studies that is due to heterogeneity rather than chance [62,63]. Funnel plots were also reviewed.

Mean FRAA titers were pooled across studies using standard methodology [64]. To compare FRAA titers across groups, pooled Cohen’s d’ (a measure of effect size) was calculated from the standardized mean difference of outcome measures using the inverse variance heterogeneity model, since it has been shown to resolve issues with underestimation of the statistical error and spuriously overconfident estimates with the random effects model when analyzing continuous outcome measures [65].

The outcome measures used across treatment studies were different in most cases, making a formal meta-analysis of any particular outcome not possible. Thus, the effect size, as measured by Cohen’s d’, was calculated where possible so the strengths of effects could be compared across studies. For controlled studies, the effect size represented the difference between the treatment and the control groups, whereas for uncontrolled studies the effect size was calculated only for the treatment group. Only a subset of studies contained the information needed to calculate the effect size. For example, for the calculation of effect size, the change in the outcome needed to be reported across the treatment period; reported mean values of the outcome before and after treatment were insufficient to calculate an effect size. Effect sizes were considered small if Cohen’s d’ was 0.2; medium for Cohen’s d’ was 0.5; and large if Cohen’s d’ was 0.8 [66].

## 3. Results

This section will first discuss folate pathway abnormalities related to ASD, followed by treatment of folate pathway abnormalities.

### 3.1. Central Folate Pathway Abnormalities and the Folate Receptor Alpha Autoantibody

#### 3.1.1. ASD Prevalence in CFD

Five case-series [20,21,33,35,67] described 79 children with CFD in which ASD was assessed (Appendix A) resulting in a prevalence of 44% (21%, 70%) of ASD in CFD (Table 1). Removing the one study with a very high prevalence rate because of asymmetry [33] lowered the pooled prevalence rate to 32% (19%, 45%).

#### 3.1.2. Cerebral Folate Deficiency in Autism Spectrum Disorder

Two case-series [34,68], two case-reports [36,69] and four prospective cohort-studies [37,45,70,71] described 172 individuals with idiopathic ASD who had CSF measurements (See Appendix A). The pooled prevalence of CFD in ASD was 38% (11%, 71%) with a significant variation across studies due to heterogeneity driven by three studies with very high prevalence rates [34,36,68] and four studies with very low prevalence rates [37,45,70,71]. Two studies with high prevalence rates reported severe patients; one study was an older case series specifically examining low-functioning ASD with neurological deficits [34]; and one was a case report of a child with mental retardation and seizures [36]. Two case series examined patients with mitochondrial disorders with very different prevalence; the series which reported a new type of non-traditional mitochondrial disorder reported a high prevalence (100%) of CFD [68], while the series which reported classical mitochondrial disorders reported a low (5%) prevalence of CFD [71]. The overall pooled prevalence was 43% (0%, 100%) with significant heterogeneity. One study found no correlation between CSF levels of 5-MTHF and measures of autism symptomatology as measured by the Autism Diagnostic Observation Schedule (ADOS) calibrated severity score, adaptive behavior as measured by the Vineland Adaptive Behavior Scale (VABS), and cognitive functioning as measured by the Mullen Scales of Early Learning [70].

#### 3.1.3. Prevalence of Autoantibodies to the Folate Receptor Alpha in ASD

Nine studies examined the prevalence of FRAAs in ASD [37,38,39,40,41,42,44,45,72]. Two sets of studies [37,39] and [40,41,72] reported on the same cohort of patients. Additionally, one study reported the prevalence in two subsets of patients; those treated with *d,l*-leucovorin (*n* = 82) and those untreated (*n* = 84) [44]. This resulted in six unique studies that examined the prevalence of FRAAs in children with ASD (See Appendix A).

The correlation between FRAAs and patient characteristics have been outlined in several studies. Two studies [35,37] reported a significant inverse correlation between higher blocking FRAA serum titers and lower 5-MTHF CSF concentrations. One study found that blocking FRAA decreased with age [37] with another study reporting increased blocking FRAA over a 2-year period with continued use of cow’s milk [35]. One study suggested that children with ASD were significantly different in physiological and developmental characteristics depending on whether they had the blocking or binding FRAA; the binding FRAA was associated with higher serum B12 concentration, while the blocking FRAA was associated with better redox metabolism, inflammation markers, communication on the VABS, stereotyped behavior on the Aberrant Behavioral Checklist (ABC) and mannerisms on the Social Responsiveness Scale (SRS) [40]. In another study, FRAA positive children were more likely to have a medical diagnosis of hypothyroidism [37]. Two studies found a positive correlation between the blocking FRAA titers and thyroid stimulating hormone (TSH) concentrations [39,41]. One study examined this relationship in detail, finding that thyroid hormone was rarely outside the normal range, suggesting that the relationship between TSH and thyroid hormone levels were altered in some children with ASD. Interestingly, this study also found that FRAAs bind to prenatal thyroid tissue in early gestation (prior to 18 weeks) suggesting that FRAAs during gestation could affect the programing of the hypothalamic-pituitar*y*-axis regulation of thyroid hormone [37].

Five studies [37,38,40,42,45] from three sets of investigators found a blocking FRAA prevalence of 46% (27%, 64%) in ASD with a significant heterogeneity but no asymmetry, indicating variation in the underlying ASD samples across studies. Four studies [37,40,42,45] from two sets of investigators found a binding FRAA prevalence of 49% (43%, 55%) without significant variation across studies. Five studies [37,40,42,44,45] from three sets of investigators found an overall FRAA prevalence of 71% (64%, 77%) without significant variation across studies. From the three studies reporting both FRAA titers [37,40,42] and one study only measuring blocking FRAAs [38] in ASD, pooled mean blocking FRAA titers was 0.85 pmol of IgG antibody per ml of serum (95% CI: 0.59, 1.11) and binding FRAA was 0.42 pmol of IgG antibody per ml of serum (95% CI: 0.35, 0.49).

One study that looked at the blocking FRAA over a five-week period found that in four patients with predominately negative titers, if tested over several weeks, they may have low positive (~0.3–0.4 pmol of IgG antibody per ml of serum) at some point, while others with the most high titers can have low or negative titers at some time points [45].

Four studies [37,38,42,44] from three sets of investigators examined blocking and binding FRAA prevalence in parents and TD siblings of children with ASD. Prevalence of the blocking, binding and either FRAA was 30% (19%, 44%), 23% (0%, 61%) and 45% (27%, 60%) in parents of children with ASD, respectively. All studies demonstrated significant heterogeneity without asymmetry suggesting variation in the underlying ASD samples across studies. Interestingly, TD siblings of children with ASD appear to have a similar prevalence as the children with ASD themselves with a pooled prevalence of 38% (19%, 58%), 40% (9%, 77%) and 61% (28%, 97%) for blocking, binding and either FRAA, respectively, without significant variation across studies.

The prevalence of FRAAs in TD children without ASD siblings was assessed in two studies [42,45] with a pooled prevalence of 4% (1%, 10%), 10% (10%, 48%) and 15% (0%, 46%) for blocking, binding and either FRAA, respectively; this is much lower than the FRAA prevalence in children with ASD or their siblings. However, there was significant variability in the binding FRAA across these two studies, demonstrating the need for larger cohorts of non-sibling control samples.

One study [38] examined the prevalence of the blocking FRAA in developmentally delayed children without ASD and found a pooled prevalence of 5% (0%, 14%); this is much lower than the FRAAs prevalence in children with ASD or their TD siblings.

#### 3.1.4. Comparison of Prevalence of Autoantibodies to the Folate Receptor Alpha in ASD to Other Groups

A meta-analysis was used to calculate the odds ratio of having FRAAs in children with ASD compared to various groups. Five studies [37,38,42,44,45] reported FRAAs in both children with ASD and their parents. The odds of being positive for the blocking or either FRAA, but not binding alone, was significantly increased in children with ASD as compared to their parents (Table 2). The odds of being positive for the FRAA was not different between children with ASD and their TD siblings. However, children with ASD demonstrated a significantly increased odds of being positive for the blocking, binding and either FRAA as compared to TD children without an ASD sibling, and as compared to developmentally delayed children without ASD for the blocking FRAA.

Two studies [37,42] compared the serum concentrations of FRAAs in children with ASD compared to parents and/or TD siblings while one study compared FRAA in children with ASD to TD children without ASD siblings [42]. Meta-analysis found that the mean blocking FRAA titer in ASD was significantly higher than parents (d’ = 0.26 (0.116, 0.36), *p* < 0.0001) and siblings (d’ = 0.29 (0.15, 0.43), *p* < 0.0001) with small-to-medium effect sizes and significantly higher than controls with a very large effect size (d’ = 2.93 (1.85, 4.01), *p* < 0.0001). However, the mean binding FRAA titer in ASD was significantly higher than in parents (d’ = 0.14 (0.06, 0.22), *p* < 0.001) but not siblings (d’ = 0.06 (−0.04, 0.17), *p* = n.s.) with small effect sizes and significantly higher than controls but with a small effect size (d’ = 0.16 (0.07, 0.25), *p* < 0.001).

Finally, in one study that measured blocking FRAAs in children with CFD with and without ASD there was no significant difference between groups (Mean (SD) 1.17 (0.84) pmol/mL and 1.78 (1.99) pmol/mL, respectively, t(23) = 0.91, *p* = 0.37) [35].

### 3.2. Treatment of ASD with d,l-Leucovorin

As seen in Figure 1, 20 studies were identified which studied *d,l-*leucovorin treatment in individuals with ASD including four placebo-controlled studies [72,73,74,75], three prospective, controlled studies [37,44,76], nine prospective studies without a control group [20,21,33,34,35,77,78,79,80] (two studies examined the same cohort of patients [78,79]), and four case reports/series [36,67,68,69].

A review of these studies on treatment of children with ASD with *d,l*-leucovorin appears to fall into three categories. Firstly, children with ASD and concomitant CFD were studied. Secondly, *d,l*-leucovorin was studied in isolation for treating idiopathic ASD. Thirdly, some studies used *d,l*-leucovorin in combination with other nutritional supplements to treat ASD symptoms. Each of these approaches to using *d,l*-leucovorin is outlined in separate sections below.

#### 3.2.1. Treatment with *d*,*l*-Leucovorin in ASD and Comorbid CFD

Nine unique case-series/reports describe children with ASD and comorbid CFD treated with *d,l*-leucovorin (See Appendix A) [20,21,33,34,35,36,67,68,69]. Two studies, one [36] case report and one case series [33], reported on the same child. A meta-analysis was conducted to determine the prevalence of improvement in symptoms as a result of *d,l*-leucovorin treatment in children with CFD with and without ASD (See Table 3; Appendix A). Six studies [21,33,34,35,67,69] reported a response rate of 67% for improvement in ASD symptoms with *d,l*-leucovorin treatment. Response to *d,l*-leucovorin in irritability was studied in children with ASD in three studies [21,34,35] and in children without ASD in three studies [21,33,35]. For those with ASD, irritability improved in 58% while in those without ASD irritability improved in 47% with a wide variation among studies, because one study demonstrated a high response rate of 88% [21] while the other two studies demonstrated much lower response rates of 22% [35] and 0% [33].

Five studies examined ataxia in ASD [21,33,34,35,67] while two studies examined ataxia in children without ASD [21,35] with both groups of children demonstrating a high rate of response of ataxia to *d,l*-leucovorin treatment (88% and 72%, respectively). Pyramidal signs were reported in four studies for those with ASD [21,34,35,67] and in two studies for those without ASD [21,35]. Response was relatively high for those with ASD (76%) while relatively low for those without ASD (33%), although there was wide variation in response rates across studies in both groups.

The response of dyskinesias and other movement disorders to *d,l*-leucovorin treatment was examined in ASD in four studies [21,33,34,35] and in children without ASD in three studies [21,33,35]. Movement disorders improved with *d,l*-leucovorin in 47% of those with ASD while the response rate was much lower for those without ASD (18%).

Six studies examined epilepsy response for those with ASD [21,33,34,35,67,69] while four studies examined response to epilepsy for those without ASD [21,33,35,67]. Epilepsy improved in 75% of children with ASD, but the response rate was somewhat lower (54%) and much more variable for those without ASD since there was a large variation across studies, perhaps driven by the overall small number of cases.

Treatment with *d,l-*leucovorin was reported to improve CSF 5-MTHF concentrations in several studies. One year of 0.5–1 mg/kg/day *d,l-*leucovorin treatment normalized 5-MTHF CSF concentrations in 90% of children in one case-series [20]; 0.5–1 mg/kg/day *d,l-*leucovorin improved CSF 5-MTHF concentrations in seven children in another study [35]; *d,l-*leucovorin 1–3 mg/kg/day over a 12-month period led to improvements in CSF 5-MTHF concentrations in 21 patients in a third study [34]; *d,l*-leucovorin, 0.5–9.0 mg/kg/day orally, followed by *d,l*-leucovorin, 6 mg/kg Q6 h for 1d IV monthly for 6 months normalized CSF 5-MTHF in a fourth study [69]. Additional treatments which resulted in clinical improvements included the removal of cow’s milk in one study [35]. One study reported a trend towards more robust improvements in younger children as compared to older children [34].

#### 3.2.2. Treatment with *d*,*l*-Leucovorin in General ASD: Leucovorin Only

Appendix A lists the five studies [37,72,74,80,81] performed by three different sets of investigators which have examined the use of *d,l-*leucovorin in children with idiopathic ASD without additional treatments in order to determine if *d,l-*leucovorin administered by itself is a useful treatment for ASD. Table 4 outlines the effect sizes of some of the key outcome measures used in these studies.

In a medium-sized (*n* = 44) open-label, prospective, controlled study which used a wait-list control group that did not receive any new interventions, children with ASD who were known to be positive for a FRAA were treated with 2 mg/kg/day (max 50 mg per day) of *d,l-*leucovorin over a mean period of 4 months [37]. Using the Parent Rated Autism Symptomatic Change Scale, significant improvements were reported in verbal communication, expressive and receptive language, attention and stereotypy with mostly large effect sizes (See Table 4). Interestingly, improvement in verbal communication and expression language demonstrated greater improvement as age increased in children who were negative for the binding FRAA but demonstrated lesser improvement as age increased for children who were positive for the binding FRAA.

One study reported nonsignificant improvements with *d,l*-leucovorin in 12 patients with ASD using 2 mg/kg/day (max 50 mg/day) of *d,l*-leucovorin over 12 weeks. Nonsignificant improvements were observed on the ABC (2.4-point improvement) and SRS (7.8-point improvement) while a 0.8-point nonsignificant worsening on the PedsQL was also observed. Urinary metabolites showed changes during the study including a 24.8-fold increase in 5MTHF concentrations; a 10.1-fold increase in 1-stearoyl-2-arachidonoyl-GPC; a 9.3-fold increase in 1-stearoyl-2-oleoylGPC; a 9.2-fold increase in alpha-tocopherol; and a 7.8-fold increase in 1-stearoyl-2linoleoyl-GPC (7.8); however, the authors noted that the study lacked to power to determine if changes in urinary metabolites predicted treatment response [80].

Two placebo-controlled studies of *d,l*-leucovorin used *d,l*-leucovorin in children with idiopathic ASD without additional treatments. In a medium-sized (*n* = 48) double-blind, placebo controlled (DBPC) study of 48 children with ASD without known CFD, 23 children received *d,l*-leucovorin calcium (2 mg/kg/day; maximum 50 mg/day) and 25 received a placebo [72]. Significant improvements were seen in the primary outcome measure of verbal communication with an overall medium-to-large effect size and a larger effect size for those positive for at least one FRAA (Table 4). The primary outcome measure exceeded the minimal clinically important difference defined as a change of five standardized points on the language assessments over 3 months. Improvements were also observed in the secondary outcome measures of the VABS daily living skills, ABC irritability, social withdrawal, stereotypy, hyperactivity and inappropriate speech; and in the Autism Symptom Questionnaire (ASQ) stereotypic behavior and total score. ABC social withdrawal, stereotypy, and inappropriate speech and ASQ stereotypic behavior and total score exceeded the predefined minimal clinically important difference. The number needed to treat (NNT) for improvements in verbal communication was 2.4 in all treated children, and 1.8 in children who were positive for at least one FRAA [72].

The second placebo-controlled study was a smaller (*n* = 19) single-blind, placebo-controlled study of 19 children with ASD; 9 children received *d,l-*leucovorin (5 mg twice daily; 0.29–0.63 mg/kg/day) for 12 weeks and 10 children received a placebo [74]. Significant improvements were found in ADOS total score and social interaction subscale with large to very large effect sizes and in the communication subscale with a medium effect size (See Table 4). These changes were significant in the treated group but not in the placebo-control group. The SRS, completed by the parents, showed nonsignificant improvement with a very small effect size.

In a retrospective national survey of 1286 participants with ASD or their parents/caregivers, a number of nutritional supplements were rated for changes in behaviors and AEs. Higher dose folinic acid (more than 5 mg/day orally) improved cognition in 33%, attention in 29%, and language/communication in 24%. A moderate dose of folinic acid (below 5 mg/day orally) improved language/communication (20%). The overall adverse effect rating was minimal [81].

These series of studies provide evidence that *d,l-*leucovorin is helpful for a wide variety of core and associated ASD symptoms. The three sets of investigators used very different doses of *d,l-*leucovorin. Three studies [37,72,80] used a relatively high dose (2 mg/kg/day; maximum 50 mg/day) while one study [74] used a much lower dose (5 mg twice daily; 0.29–0.63 mg/kg/day) but both dosing parameters were associated with significant clinical improvements.

#### 3.2.3. Treatment with *d,l*-Leucovorin in General ASD: Combined with Other Supplements

Six studies (spanning seven reports) from four sets of investigators used *d,l-*leucovorin in combination with nutritional supplements or other medications to treat ASD (Appendix A). The most recent study added *d,l*-leucovorin or placebo to risperidone in a DBPC study [75]. One set of investigators examined a multivitamin-mineral complex (MVMC) in two controlled studies [73,76] (one study was controlled with a placebo group and one utilized untreated children as controls), while another set of investigators examined *d,l-*leucovorin along with other indicated treatments using a prospective clinical protocol with a control group of ASD children receiving only standard behavioral and educational therapy without medical interventions [44]. Finally, one set of investigators examined *d,l-*leucovorin along with other treatments in an open-label fashion without comparison groups [77,78,79]. All of these studies were performed on children with ASD without known CFD.

In a medium-size (*n* = 55) DBPC study, children received either *d,l-*leucovorin (2 mg/kg/day up to 50 mg daily) or placebo in two divided daily doses along with risperidone for 10 weeks [75]. Risperidone was started at 0.5 mg and increased by 0.5 mg weekly up to 1 mg for children <20 kg and 2 mg for children ≥20 kg. The ABC inappropriate speech was the primary outcome measure with the remainder of the ABC subscales as the secondary outcome measures. All ABC subscales improved more in the leucovorin group as compared to the placebo group with statistical significance in all subscales except for social withdrawal. The authors used two different measures of effect size Cohen d’ and η^2^ which provided different estimates of the effect sizes with η^2^ showing medium effect sizes and Cohen d’ demonstrating extremely large effect sizes. Due to this discrepancy, these results are not listed in Table 4 because of their ambiguity.

In a large (*n* = 141) DBPC study, children with ASD were treated with either a MVMC which contained 550 µg of *d,l-*leucovorin (*n* = 72) or a placebo (*n* = 69) for 3 months. The Parental Global Impressions-Revised (PGI-R) demonstrated improvements in communication and behavior (See Table 4). In addition to the *d,l-*leucovorin, the active treatment contained vitamins A, C, D3, E, K, B1-B6, B12, folic acid, biotin, choline, inositol, mixed carotenoids, mixed tocopherols, CoEnzyme Q10, N-acetylcysteine, calcium, chromium, copper, iodine, iron, lithium, magnesium, manganese, molybdenum, phosphorus, potassium, selenium, sulfur and zinc [73].

In another medium-sized (*n* = 67) prospective, open-label, controlled study of 67 individuals with ASD, a MVMC containing 600 µg folate mixture (*d,l-*leucovorin, folic acid and 5-MTHF combined) per day was given to 37 individuals with ASD for 12 months, while 30 individuals were untreated [76]. Improvements were found in several of the Parental Global Impressions-2 (PGI-2) measures (Table 4) as well as in the Childhood Autism Rating Scale 2 (CARS2) score, Short Sensory Profile, SRS, ABC, Autism Treatment Evaluation Checklist (ATEC) and VABS. Besides the folate mixture, the treatment contained vitamins A, C, D3, E, K, B1-B6, B12, biotin, choline, inositol, mixed tocopherols, CoEnzyme Q10, N-acetylcysteine, Acetyl-L-Carnitine, calcium, chromium, vanadium, boron, lithium, magnesium, manganese, molybdenum, potassium, selenium, sulfur and zinc.

In a large (*n* = 166) prospective, case-control, open-label study, 82 children with ASD (ages 1–15.9 years) were treated with *d,l-*leucovorin 0.5–2 mg/kg/day (maximum 50 mg/day) for 2 years and were compared to 84 untreated children with ASD (ages 1–16.8 years) who were matched for age, gender, CARS score and FRα autoantibody status [44]. The untreated control group of 84 ASD children received only standard behavioral and educational therapy without additional medical interventions. Blood tests were performed to identify nutritional deficiencies and abnormal oxidative stress biomarkers which were treated with other vitamins and minerals (such as vitamins A, C, D, E, zinc, selenium, manganese, and CoEnzyme Q10, when indicated by testing) in the treated children. *d,l-*Leucovorin was given to the children who had positive FRAAs (62 of 82 children with ASD had FRAAs, 75.6%). The other 20 children without positive FRAAs did not receive *d,l-*leucovorin. This study reported improvements in mean CARS scores from severe ASD (mean (SD): 41.34 (6.47)) to mild or moderate ASD (mean (SD): 34.35 (6.25)) in all age cohorts of the treated group (See Table 4). In the untreated group of 84 children, there was no significant change in the mean CARS score over the study period. The authors reported “complete recovery” in 17 of 82 children (21%).

Two of the open-label, prospective studies from one set of investigators examined the use of *d,l-*leucovorin in a cohort of individuals with ASD without a control group. In the first study which included eight children with ASD, 800 µg of *d,l-*leucovorin and 1000 mg of betaine was given twice a day for 4 months; significant improvements (*p* ≤ 0.05) were found in the concentrations of methionine, S-adenosylmethionine (SAM), homocysteine, cystathionine, cysteine, total glutathione (tGSH) concentrations, SAM:S-adenosylhomocysteine (SAH) and tGSH:GSSG; clinical improvements in speech and cognition were noted by the attending physician but were not formally quantified [77]. James et al., 2009 [78] studied 40 children with ASD and administered 400 µg of *d,l-*leucovorin twice a day and 75 µg/kg methyl-cobalamin injected subcutaneously twice a week. This treatment led to significant increases in cysteine, cysteinyl-glycine, and glutathione concentrations (all *p* < 0.001); significant improvements were observed in all subscales of the VABS [79].

### 3.3. Adverse Effects Reported with d,l-Leucovorin Treatment in ASD

Overall, the placebo-controlled studies support the minimal AEs associated with leucovorin treatment. No significant difference in AE frequency as compared to placebo was reported in the single-blind, placebo-controlled study [74], in two DBPC studies [72,75], or in the last DPBC study for patients who followed the protocol [73].

To investigate the consistency of reported AEs, a meta-analysis was performed on reported AEs for patients on leucovorin treatment, separately for studies examining only leucovorin and for those studies which combined leucovorin with other supplements or treatments (Table 5). For studies which only treated with leucovorin, consistently reported AEs included excitement or agitation (11.7%), aggression (9.5%), insomnia (8.5%), increased tantrums (6.2%), headache (4.9%) and gastroesophageal reflux (2.8%). For studies which used leucovorin in combination with other agents, AEs that were consistently reported included worsening behavior (8.5%) and aggression (1.3%).Interestingly, Frye et al., 2020 [9] examined the reported targeted AE of agitation and excitability every 3 weeks during their previous 12-week study of 2018 [72]. This AE was reported at almost the exact same frequency in the treatment and placebo group until the 9th week of treatment when it precipitously dropped in frequency in the treatment, but not the placebo, group, demonstrating the improvement of this reported AE with longer exposure to the medication.

## 4. Discussion

This systemic review found CFD is associated with ASD. One cause of CFD is FRAAs which are a common finding in children with ASD. *d,l*-Leucovorin is a proven treatment for CFD that has been studied in ASD and can normalize 5-MTHF concentrations in the CSF.

The meta-analysis found a pooled prevalence of ASD in CFD of 44%. The pooled prevalence of CFD in ASD was 38% with the etiology attributed to FRAAs in 83% of the cases. The pooled prevalence of blocking, binding and either FRAA in idiopathic ASD was 46%, 49% and 71%, respectively. Children with ASD were more likely than their parents to have blocking or at least one FRAA but were not more likely than typically developing (TD) siblings to have FRAAs. For those with ASD, blocking FRAA titers were significantly higher than their parents or TD siblings, while binding FRAA titers were significantly higher than parents but not TD siblings. Children with ASD were more likely to have positive FRAAs as compared to non-related TD children or children with developmental delay without ASD. Children with ASD demonstrated significantly higher blocking and binding FRAA titers than normal controls with the effect size for blocking FRAAs being very large. FRAAs, particularly blocking FRAAs, are highly prevalent in children with ASD, and may serve as a biomarker for treatment.

This systemic review identified 20 studies which described treating individuals with ASD using *d,l*-leucovorin with a dose typically ranging from 0.5 to 2.5 mg/kg/day. For children with ASD and CFD, *d,l*-leucovorin was particularly effective (>75% response rate) for treating ataxia, pyramidal signs and epilepsy, although it also improved ASD symptoms, irritability and movement disorders in eight case-series. In three controlled studies, *d,l*-leucovorin alone was found to consistently improve communication with medium-to-large effect sizes, but also was shown to have a positive effect on core ASD symptoms and associated behaviors (attention and stereotypy) in individual studies with large effect sizes. In five controlled and uncontrolled studies, *d,l*-leucovorin in combination with vitamin and/or mineral supplements was found to significantly improve core ASD symptoms, communication, behavior and associated symptoms with medium-to-large effect sizes. This systemic review found *d,l*-leucovorin is associated with improvements in core and associated symptoms of ASD with the strongest evidence coming from the blinded, placebo-controlled studies. Most studies reported mild to no AEs, and AEs in the placebo-controlled studies were similar in treated and untreated individuals.

### 4.1. Dosing of d,l-Leucovorin in ASD

Most studies used 0.5 to 2.5 mg/kg/day of oral *d,l*-leucovorin but one study reported using up to 9 mg/kg/day in a child with ASD and then added 24 mg/kg/day IV (divided into 4 doses) for one day every month for 6 months with a decrease in severity and seizures along with improved eye contact [69]. Therefore, higher doses of *d,l*-leucovorin appear necessary in order to achieve a higher brain folate level and clinical improvements.

### 4.2. Time Period Needed for Maximal d,l-Leucovorin Treatment Effects

One case report of a child with CFD and mitochondrial disease (this child did not have ASD) reported improvements with *d,l*-leucovorin over a 3 year period [26]. Other studies in ASD reported significant improvements over periods of 1 year [34] to 2 years [35,44]. However, other much shorter studies also demonstrated significant improvement in ASD symptoms [37,72,74]. Therefore, although some individuals might show a relatively quick response to *d,l*-leucovorin, it may take 1–2 years to observe maximal clinical improvements.

### 4.3. The Effect of d,l-Leucovorin on the Core Symptoms of ASD

Some of the improvements with *d,l*-leucovorin in the reviewed studies were in core ASD symptoms (communication and repetitive and stereotyped behavior). To date, there are no FDA approved medications available to treat the core symptoms of ASD and the only two currently approved medications for ASD (aripiprazole and risperidone) are only approved for treating irritability associated with ASD, which is not considered a core symptom of ASD. In addition, aripiprazole and risperidone have been shown in repeated studies to potentially cause long-term metabolic and neurological adverse effects [82]. Therefore, *d,l*-leucovorin is especially promising since many of the reviewed studies found improvements in core ASD symptomology. *d,l*-Leucovorin also has a much better safety profile and less AEs compared to aripiprazole and risperidone.

### 4.4. Seizures and Treatment with d,l-Leucovorin

Six studies reported reductions in seizures in children with ASD and CFD using *d,l*-leucovorin [20,33,34,67,69] even in patients with difficult-to-control seizures [20,33]. One child had a breakthrough seizure with discontinuation of *d,l*-leucovorin for 2 weeks [67]. It is possible that epilepsy in these children could be caused by CFD or FRAAs [83]. This is potentially an important finding as many treatments for epilepsy in children treat the seizure condition but not the potential underlying cause or contributing factor. Of note, an animal model reported that certain antiepileptic medications might disrupt folate transportation into the CNS [84]. More studies would be helpful in determining if CFD or FRAAs are an underlying cause of seizures in children with ASD and if treatment with *d,l*-leucovorin is a useful medication for mitigating seizures in these children.

### 4.5. Treatment of d,l-Leucovorin in Patients with Mitochondrial Dysfunction and ASD

Mitochondrial dysfunction is a common comorbidity in ASD, with studies reporting 30–50% of individuals with ASD possessing biomarkers of mitochondrial dysfunction [6,46] and up to 80% having abnormal electron transport chain activity in immune cells [47,48]. Several studies have linked CFD to mitochondrial disease in individuals without ASD [24,25,26,27] and mitochondrial dysfunction in children with ASD [68]. In children with ASD and CFD, meta-analysis revealed a prevalence of mitochondrial dysfunction of 43% as a potential etiology of CFD. Thus, even in the absence of FRAAs, mitochondrial disease and dysfunction should also be considered as a potential cause of CFD in individuals with ASD.

Treatment of mitochondrial dysfunction with mitochondrial-related cofactors and vitamins, including carnitine [85,86], ubiquinol [87] and a “mitochondrial cocktail” containing carnitine, CoEnzyme Q10 and Alpha-Lipoic Acid [88] has been reported to improve some symptoms of ASD. Folate has also been reported to increase ETC Complex I activity in children with ASD and mitochondrial disease and positively modulate the coupling of ETC Complex I and IV and ETC Complex I and Citrate Synthase [89]. *d,l*-Leucovorin rapidly accumulates in mitochondria [90] and is the preferred form of folate in treating mitochondrial dysfunction [91]. *d,l*-Leucovorin has been reported to improve mitochondrial related symptoms and laboratory findings in some patients with mitochondrial disease in doses ranging from 1–8 mg/kg/day [24,26,27,92] including in one child with ASD [68]. Since mitochondrial dysfunction is relatively common in individuals with ASD [6,46,47,48,93] and *d,l*-leucovorin appears to help patients with mitochondrial dysfunction [24,26,27,92], one mechanism by which *d,l*-leucovorin might help improve ASD symptoms is by improvements in mitochondrial function.

### 4.6. Safety of d,l-Leucovorin in ASD

*d,l*-Leucovorin was first approved in the United States in the 1950s and has been used continuously since then to reduce toxicities associated with folate pathway antagonists. Therefore, it has a strong and long track record of safety. Most of the reviewed studies reported mild to no AEs with *d,l*-leucovorin. Some studies reported mild behavioral problems, diarrhea/constipation, and aggressive behaviors. In the placebo-controlled studies, AEs were similar in treated and untreated individuals. Two studies used *d,l*-leucovorin for up to 2 years [35,44] without significant AEs. Two other studies used *d,l*-leucovorin for one year without significant AEs [20,34]. Therefore, the use of *d,l*-leucovorin appears to be safe for at least 2 years of use in most individuals with ASD.

### 4.7. Screening for FRα Autoantibodies in ASD

The meta-analysis reported a pooled prevalence of a positive FRAA in children with ASD and concomitant CFD [21,34,35,68] of 83% (69%, 94%) with consistency across studies. Two studies [35,37] reported a significant correlation between higher blocking FRAA concentration and lower CSF levels of 5-MTHF. One study reported that children with ASD who had higher titers of FRAAs had less robust improvements [44]. Only two studies reported one child each with a mutation in the FOLR1 gene which could account for CFD [67,69]. These findings suggest that FRAAs are the major cause of CFD in children with ASD. One set of authors suggested screening young children and infants who have developmental delay and ASD features for FRAAs and starting treatment as soon as possible with *d,l*-leucovorin, especially since younger children generally show more robust improvements [34]. Some patients have intermittently positive FRAA levels and may need to be tested several times if they are negative [45]. Another approach which has been suggested by some authors is an empiric trial of *d,l*-leucovorin in children with ASD without performing a lumbar puncture to confirm CFD, especially given the excellent safety profile of *d,l*-leucovorin [94]. Additionally, one study reported improvements with *d,l*-leucovorin in children with ASD who did not have known CFD and did not possess positive FRAAs [72], further suggesting that empiric treatment in individuals with ASD with *d,l*-leucovorin is a reasonable approach.

Evidence has linked the FRAAs with 5-MTHF concentrations in the CSF in ASD and demonstrated that they can predict response to *d,l*-leucovorin treatment, suggesting that FRAAs are involved in the disruption of folate metabolism in patients with ASD. Additionally, animal models have validated their pathophysiological mechanism [95,96]. However, the meta-analysis suggested a high rate of FRAAs not only in children with ASD but also in their parents and TD siblings, but not in unrelated TD controls or children with developmental delay without ASD. This suggests that FRAA may be one of several mechanisms involved in the disruption of folate metabolism that can contribute to CNS folate disruption. Other factors may be involved. For example, if a child with ASD has FRAAs and other medical comorbidities such as mitochondrial dysfunction, this might explain why a sibling (without these medical comorbidities) can have a positive FRAA but not develop ASD. Polymorphisms in folate genes are also overrepresented in children with ASD and their mothers, suggesting that FRAAs are not alone is disrupting folate metabolism. It is possible that combinations of several mechanisms involved in disrupting folate metabolism many be needed to lead to enough disruption in neurodevelopment to lead to ASD. Timing may also be an important factor as FRAAs and CFD have been related to other psychiatric disorders when they occur outside of childhood such as in schizophrenia [53] and depression [97]. Of note, none of the reviewed studies examined a potential correlation between ASD severity and FRAA concentrations. In the future, studies that examine this would be useful.

### 4.8. Adjunctive Treatments Studied for FRAA Positive Patients

Cow’s milk appears to regulate FRAA titers; 6 months of a milk-free diet resulted in a significant decrease in FRAA titers with re-exposure to cow’s milk increasing this titer, with the titer often rising above the original titer level before initially discontinuing cow’s milk [35]. Concomitant with the decrease in the FRAA titers, patients with CFD who went on a cow’s milk free diet demonstrated improvements in ataxia, improved seizure control, and improved ASD symptoms. This is believed to occur because milk contains the FRα protein which may react immunologically in the gut or cause an increase in cross-reactive FRAAs in the blood. Several types of diets (such as a casein-free diet) which have been reported to show some effectiveness in ASD are milk-free diets [98,99] although this has not been found in some studies [100]. Thus, it is possible that dietary treatments not uncommonly used to treat children with ASD may have a therapeutic effect by lowering the concentration of FRAAs in the blood. Unfortunately, large studies have not examined the potential benefit of adding a milk-free diet to *d,l*-leucovorin treatment but it is probably prudent to recommend a milk-free diet when FRAAs are present. Interestingly, in the group of patients with CFD who continued to drink bovine milk, the FRAA concentrations continued to rise over a two-year period [35]. In this study, the use of goat milk caused less elevation in the FRAA concentration and thus might be a better alternative to bovine milk [35]. It is important to recognize that this effect may not generalize to other forms of dairy that do not have intact milk proteins. Indeed, dairy is an important source of calcium, which is a critical nutrient in childhood for bone health.

### 4.9. Treatments That Support Folate Transport into the Brain

Presumably, when the FRα is partially blocked, the RFC may be the main alternative transportation mechanism of folates into the brain. In a human cerebral microvascular endothelial cell model, 1,25-dihydroxyvitamin D_3_ was shown to up-regulation RFC mRNA and protein expression through activation of the vitamin D receptor [101]. In the same cell model, nuclear respiratory factor 1 and peroxisome proliferator-activated receptor-γ coactivator-1α signaling were found to modulate RFC expression and transport activity with this pathway upregulated by treatment with pyrroloquinoline quinone (PQQ) resulting in increased RFC expression and folate transport activity [102]. Furthermore, 1,25-dihydroxyvitamin D_3_ was found to rescue CFD in a knockout FOLR1 mouse model [103].

Interestingly, meta-analysis has shown that vitamin D3 800IU to 2,000IU supplementation improves core ASD symptoms as measured by the Social Responsiveness Scale or CARS in three studies in children with ASD who were not vitamin D deficient [104] and PQQ has been shown to improve social behavior in the bilateral whisker trimming for 10 days after birth in a mouse model which is characterized by its abnormal social behavior [105]. Thus, although no studies have been conducted in ASD or CFD to determine whether these supplements may improve function in children with ASD who have FRAAs or CFD, such supplements may have some utility in these children. Future studies will need to address this possibility.

### 4.10. Adjunctive Treatments to Support Folate Metabolism

Folate is essential for several important biochemical pathways, particularly the function of methylation metabolism and the production of purines and pyrimidines. Several studies used *d,l*-leucovorin in combination with other cofactors which could support its metabolism, including methyl-cobalamin, betaine (trimethyl-glycine) and other important cofactors. Many cofactors are essential for optimal functioning of enzymes in the folate cycle; for example, methionine synthase requires cobalamin, and methylenetetrahydrofolate reductase (MTHFR) requires nicotinamide adenine dinucleotide (NAD) which can be derived from niacin. Other factors like betaine support methylation metabolism. No study has compared the specific cofactors that could best be used in conjunction with *d,l*-leucovorin, but it is likely that other cofactors may be useful to optimize folate metabolism. One intriguing possibility is that, along with a CNS folate deficiency, individuals with ASD may also have a CNS cobalamin deficiency [106], so the addition of cobalamin could be critical in some children with ASD.

### 4.11. Therapeutic Effect of d,l-Leucovorin on Neurotransmitters

Most important for neurological outcomes for those with a CNS folate deficiency is the connection between folate and the production of neurotransmitters. Being the precursor to purines, folate is essential to produce guanosine-5’-triphosphate which is the precursor of tetrahydrobiopterin (BH_4_) [107]. BH_4_ is an important cofactor for the production of the monoamine neurotransmitters serotonin, dopamine, norepinephrine and epinephrine which are essential for behavioral regulation, mood, social function and cognition. Interestingly, like folate, a central deficiency of BH_4_ is associated with ASD [108]; children with ASD respond to BH_4_ supplementation [109]; and there is evidence that BH_4_ may be transported into the brain through the FRα [110]. The connection between folate and BH_4_ can explain how *d,l*-leucovorin can normalize CSF concentrations of serotonin and dopamine in CFD patients [28].

### 4.12. Implications of FRAAs during Pregnancy

Rodent models report that exposure of dams to FRAA’s during gestation can lead to stereotypies and anxiety in offspring [95]. Further studies showed that *d,l*-leucovorin and/or dexamethasone treatment of dams exposed to FRAAs prevented cognitive, communication and learning problems in the offspring [96]. Human studies have reported an association between FRAAs and subfertility [111], neural tube defects [112], and preterm births [113]. One study found that FRAAs bind to prenatal thyroid tissue, potentially affecting its development and potentially altering hypothalamic-pituitar*y*-axis regulation of thyroid hormones in the offspring [37]. In one case report, a pregnant women with a history of multiple complications in previous pregnancies and who was positive for FRAAs was able to conceive and have a normal pregnancy and delivery with the use of *d,l*-leucovorin, a milk free diet, and a low dose of prednisone given during pregnancy [114].

### 4.13. Limitation of Published Studies

Many of the reviewed studies had important limitations. First, there is only one medium-sized [72] and one small-sized [74] blinded, placebo controlled, single center studies that examined treatment only with *d,l*-leucovorin. Thus, clearly larger, multisite trials, which are ongoing [9], are necessary to confirm previous findings. Since many of the studies used *d,l*-leucovorin in combination with other treatments [73,76,77,78,79], other treatments may have added to the effects of *d,l*-leucovorin. In addition, studies examining CFD are rather small and do not always use standardized outcome measures [20,21,33,34,35,36,67,68].

Interestingly, in the blinded controlled studies, standardized measures of language and social functioning which were obtained by objective and blinded examiners tended to have large effect sizes, whereas parent rated measures tended to have very modest effect sizes. This reflects one of the difficulties in research in ASD where the placebo effect can be large, especially as rated by parents [115]. Such large placebo effects have washed out the effect of the treatment in many studies, resulting in many failed clinical trials. Thus, one of the strengths in the currently conducted blinded trials is the use of standard objective measures of function in blinded observers in addition to parent reported measures. This is especially important when studying more mildly affected children with ASD as the placebo effect appears to be inversely proportional to the severity of the ASD symptoms, so studies with less severe children would be expected to have a larger placebo effect [116]. This effect can explain the larger effect size in parent reported outcomes in the studies which used a non-treatment comparison group [37,76] when compared to the trials which were placebo controlled [72,73,74].

## 5. Conclusions

This systematic review and meta-analysis found *d,l*-leucovorin is associated with improvements in core and associated symptoms of ASD with the strongest evidence coming from the blinded, placebo-controlled studies. FRAAs, particularly blocking FRAAs, are highly prevalent in children with ASD, and may serve as a biomarker for treatment. The high prevalence of FRAAs in families with children with ASD suggests unknown heritability mechanisms that involve additional genetic or environmental factors which contribute to the expression of ASD in those with FRAAs. *d,l*-Leucovorin is an evidence-based treatment for ASD which has significant promise and appears safe and well-tolerated. Further studies would be helpful to confirm and expand on these findings.

## Figures and Tables

**Figure 1 jpm-11-01141-f001:**
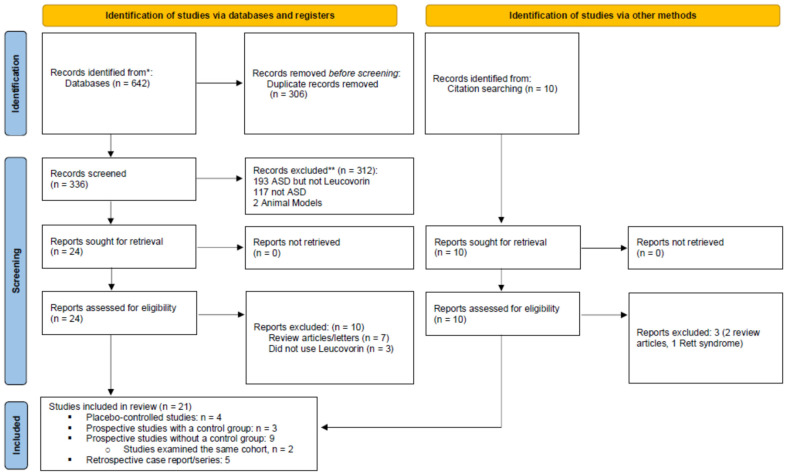
PRISMA 2020 flow diagram for this systematic review.

**Table 1 jpm-11-01141-t001:** Meta-analysis results for the prevalence of cerebral folate deficiency and folate receptor alpha autoantibodies. Pooled prevalence with 95% confidence interval, Cochran’s Q (Q), Heterogeneity Index (I^2^), Luis Furuya-Kanamori (LFK) Index and number of studies involved (N). Statistics are estimated by a random-effects model. * *p* < 0.05; ** *p* < 0.01; ^T^ Significant Asymmetry.

	Prevalence (95% CI)	Q	I^2^	LFK	N
**Cerebral Folate Deficiency**					
Prevalence of ASD in CFD	44% (21%, 70%)	16.14 **	75%	2.57 ^T^	5
Prevalence of CFD in ASD	38% (11%, 71%)	85.50 **	92%	4.20 ^T^	8
Etiology of CFD in ASD:					
Either FRα autoantibody	83% (69%, 94%)	5.19			5
Mitochondrial Dysfunction	43% (0%, 100%)	9.15 **	89%		2
Genetic abnormalities	14% (0%, 39%)	12.4 *	60%	4.63 ^T^	6
**Prevalence of FRα Autoantibody**					
Autism Spectrum Disorder (ASD)					
Blocking FRα autoantibody	46% (27%, 64%)	52.34 **	92%	0.42	5
Binding FRα autoantibody	49% (43%, 55%)	2.34			4
Either FRα autoantibody	71% (64%, 77%)	10.07			5
Parents of ASD children					
Blocking FRα autoantibody	30% (19%, 44%)	9.39 *	79%	−0.78	3
Binding FRα autoantibody	23% (0%, 61%)	13.45 **	93%		2
Either FRα autoantibody	45% (27%, 60%)	89.90 **	89%	0.05	4
Typically Developing Siblings of ASD					
Blocking FRα autoantibody	38% (19%, 58%)	1.39			2
Binding FRα autoantibody	40% (9%, 77%)	2.93			2
Either FRα autoantibody	61% (28%, 97%)	3.86			2
Typically Developing Non-sibling					
Blocking FRα autoantibody	4% (1%, 10%)	0.00			2
Binding FRα autoantibody	10% (10%, 48%)	16.33 **	94%		2
Either FRα autoantibody	15% (0%, 46%)	9.60 **	90%		2
Developmentally Delayed without ASD					
Blocking FRα autoantibody	5% (0%, 14%)				1

**Table 2 jpm-11-01141-t002:** Meta-analysis of Odds Ratios with 95% confidence interval for differences between the prevalence of Folate Receptor Alpha Autoantibodies in children with ASD to Various Comparison Groups. Odd ratios that are significant are bolded and italicized. Also listed are Cochran’s Q (Q), Heterogeneity Index (I2), Luis Furuya-Kanamori (LFK) Index and number of studies involved (N). Statistics are estimated by a random-effects model. ^Γ^ *p* < 0.05, ** *p* ≤ 0.001.

Comparison Group	Odds Ratio (95% CI)	Q	I^2^	LFK	N
Parents of ASD children					
Blocking FRα autoantibody	**2.10 (1.05, 4.21) ^Γ^**	6.37			3
Binding FRα autoantibody	3.62 (0.70, 18.84)	4.40			2
Either FRα autoantibody	**3.56 (1.62, 7.79) ****	8.60 ^Γ^	65%	−0.77	4
Typically Developing Siblings of ASD					
Blocking FRα autoantibody	2.00 (0.26, 15.21)	3.31			2
Binding FRα autoantibody	1.32 (0.44, 3.99)	1.41			2
Either FRα autoantibody	2.12 (0.40, 11.30)	3.12			2
Typically Developing Non-sibling					
Blocking FRα autoantibody	** *26.84 (7.84, 91.86) *** **	1.02			2
Binding FRα autoantibody	** *7.90 (0.70, 89.13) *** **	2.81			2
Either FRα autoantibody	** *19.03 (2.36, 153.58) *** **	3.58			2
Developmentally Delayed without ASD					
Blocking FRα autoantibody	** *25.38 (3.29, 196.02) *** **				1

**Table 3 jpm-11-01141-t003:** Meta-analysis results for the prevalence response to *d,l*-leucovorin treatment in children with CFD with and without autism spectrum disorder (ASD). Pooled prevalence with 95% confidence interval, Cochran’s Q (Q), Heterogeneity Index (I2), Luis Furuya-Kanamori (LFK) Index and number of studies involved (N). Statistics are estimated by a random-effects model. * *p* < 0.01; ^T^ Significant Asymmetry.

	Prevalence (95% CI)	Q	I^2^	LFK	N
**Children with ASD**					
Autism	67% (43%, 87%)	9.23			6
Irritability	58% (40%, 76%)	2.24			3
Ataxia	88% (75%, 97%)	3.16			5
Pyramidal Signs	76% (19%, 100%)	13.83 *	75%	−2.27 ^T^	4
Movement Disorder	47% (20%, 75%)	3.85			4
Epilepsy	75% (54%, 91%)	4.48			5
**Children without ASD**					
Irritability	47% (0%, 100%)	12.11 *	84%	−5.54 ^T^	2
Ataxia	72% (24%, 100%)	5.82			2
Pyramidal Signs	33% (0%, 100%)	13.41	92%		2
Movement Disorder	18% (1%, 46%)	2.07			3
Epilepsy	54% (0%, 100%)	12.90 *	77%	−2.31 ^T^	4

**Table 4 jpm-11-01141-t004:** Outcome measures represented in effect size in key studies which have used *d,l*-leucovorin. Cohen’s d’ was calculated for studies which provided enough information to make such calculations. For controlled studies, the effect size represented the difference between the treatment and the control group, whereas for uncontrolled studies, the effect size was calculated only for the treatment. Effect sizes were considered small if Cohen’s d’ was 0.2; medium for Cohen’s d’ of 0.5, and large if Cohen’s d’ was 0.8. Effects in bold are statistically significant.

Core ASD Symptoms	Communication	Behaviors	Other Symptoms
***d,l*-leucovorin Only Studies**			
**Frye et al., 2013 [37] (Non-Treated Wait List Controlled)**			
**Stereotypy d’ = 1.02**	**Verbal Comm d’ = 0.91** **Expressive Lang d’ = 0.81** **Receptive Lang d’ = 0.76**	**Attention d’ = 1.01** Hyperactivity d’ = 0.25	
**Frye et al., 2018 [72] (Double Blind Placebo Controlled)**			
**ABC Social Withdrawal d’ = 0.27** **ABC Stereotypy d’ = 0.60**	**Verbal Comm (All) d’ = 0.70** **Verbal Comm (FRAA+) d’ = 0.91**	**Hyperactivity d’ = 0.05**	
**Renard et al., 2020** [74] **(Single Blind Placebo Controlled)**			
**ADOS total score d’ = 1.16** **Social interaction d’ = 1.11** **SRS Total d’ = 0.03**	**Communication d’ = 0.66**		
***d,l*-leucovorin Combined with Other Supplements**			
**Adams et al., 2011 [73] (** **Double Blind Placebo Controlled** **; PGI-R Outcome)**			
Play d’ = 0.23 Sociability d’ = 0.15	Expressive Language d’ = 0.37 **Receptive Language d’ = 0.44**	**Hyperactivity d’ = 0.60** **Tantrums d’ = 0.53**	Cognition d’ = 0.34 Gastrointestinal d’ = 0.30 Sleep d’ = 0.18
**Adams et al., 2018** [76] **(Prospective Non-Treatment Controlled; PGI-2 Outcome)**			
**Play d’ = 1.50** **Sociability d’ = 1.44** **Eye Contact d’ = 1.41** **Perseveration d’ = 1.33** **Sound Sensitivity d’ = 1.14**	**Expressive Language d’ = 1.60** **Receptive Language d’ = 1.99**	**Attention d’ = 1.19** **Hyperactivity d’ = 1.46** **Tantrums d’ = 1.00** **Aggression d’ = 0.96** Self-Injury d’ = 0.69	**Cognition d’ = 1.49** **Gastrointestinal d’ = 2.09** **Sleep d’ = 0.92** **Mood d’ = 1.58** **Anxiety d’ = 0.95**
**Ramaekers et al., 2019** [44] **(Baseline Controlled)**			
**CARS d’ = 1.01–1.32**			
**Frye et al., 2013** [79] **(Baseline Controlled)**			
**Interpersonal d’ = 0.43** **Play d’ = 0.59** **Coping d’ = 0.66**	**Expressive Language d’ = 0.59** **Receptive Language d’ = 0.97** **Written Language d’ = 0.56**		**Personal d’ = 0.65** **Domestic d’ = 0.37** **Community d’ = 0.52**

**Table 5 jpm-11-01141-t005:** Meta-analysis of Adverse Effects Associated with Leucovorin in Children with ASD. Bold and italics indicate significant effects across studies.

Leucovorin Alone		Leucovorin Combination	
Adverse Effect	Incidence (95% CI)	Adverse Effect	Incidence (95% CI)
Abdominal Pain	1.7% (0.0%, 4.8%)	Abdominal Pain	2.1% (0.0%, 7.2%)
** *Aggression* **	** *9.5% (4.2%, 16.3%)* **	** *Aggression* **	** *1.3% (0.1%, 3.6%)* **
Blood in Stool	1.7% (0.0%, 4.8%)	Attention Problems	0.9% (0.0%, 2.9%)
Confusion	1.8% (0.0%, 5.0%)	Constipation/Diarrhea	7.4% (0.0%, 21.5%)
Constipation	2.6% (0.0%, 7.4%)	Dizziness	2.1% (0.0%, 7.2%)
Decreased Appetite	2.6% (0.0%, 7.4%)	Headaches	0.9% (0.0%, 2.5%)
Depression	230% (0.0%, 6.9%)	Hyperactivity	2.5% (0.0%, 8.8%)
Diarrhea	2.3% (0.0%, 7.0%)	Impulsivity	1.0% (0.0%, 2.7%)
Dry Mouth, Excessive Thirst	5.0% (0.0%, 13.6%)	Increased Appetite	3.2% (0.0%, 12.1%)
Emotional Lability	2.0% (0.0%, 7.4%)	Irritability	1.0% (0.0%, 2.7%)
** *Excitement or Agitation* **	** *11.7% (1.1%, 28.8%)* **	Nausea/Vomiting	0.9% (0.0%, 2.9%)
Gastroesophageal Reflux	** *2.8% (0.2%, 7.5%)* **	Rash	0.9% (0.0%, 2.5%)
** *Headache* **	** *4.9% (1.3%, 10.5%)* **	Reduced Sleep	1.9% (0.0%, 6.3%)
** *Insomnia* **	** *8.5% (0.2%, 23.8%)* **	Sedation	1.7% (0.0%, 5.6%)
Increased Motor Activity	7.4% (0.0%, 21.8%)	** *Worsening Behavior* **	** *8.5% (3.9%, 14.6%)* **
** *Increased Tantrums* **	** *6.2% (1.5%, 13.3%)* **		
Involuntary Movements	2.6% (0.0%, 7.4%)		
Restlessness	3.4% (0.0%, 10.6%)		
Stiffness	1.7% (0.0%, 5.0%)		
Viral Infection	10.3% (0.0%, 33.8%)		
Weight Gain	2.6% (0.0%, 7.4%)		

## Data Availability

All data are presented within the article.

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
