# Peer review of "Cerebral Folate Deficiency, Folate Receptor Alpha Autoantibodies and Leucovorin (Folinic Acid) Treatment in Autism Spectrum Disorders: A Systematic Review and Meta-Analysis"

_jpm, 2021, doi:10.3390/jpm11111141_

Round 1
Reviewer 1 Report
The Review article "Cerebral Folate Deficiency, Folate Receptor Alpha Autoantibodies and Leucovorin (Folinic Acid) treatment in Autism Spectrum Disorders: A systematic review and meta-analysis" is a very comprehensive and well researched study. There are two minor comments:
1) In the Introduction section, it would be nice if authors added details about the first identified CFD case. The details should include how the case was diagnosed. Was there a genetic testing involved?
2) In the introduction, the authors should explain (if known) how in normal developing brain, RFC and FRalpha work together. How is the balance of reduced folate maintained.
Author Response
Reviewer #1:
The Review article "Cerebral Folate Deficiency, Folate Receptor Alpha Autoantibodies and Leucovorin (Folinic Acid) treatment in Autism Spectrum Disorders: A systematic review and meta-analysis" is a very comprehensive and well researched study. There are two minor comments:
We thank the reviewer for their comments and review.
1) In the Introduction section, it would be nice if authors added details about the first identified CFD case. The details should include how the case was diagnosed. Was there a genetic testing involved?
We have expanded on the identification of the first patients with CFD in the introduction.
2) In the introduction, the authors should explain (if known) how in normal developing brain, RFC and FRalpha work together. How is the balance of reduced folate maintained.
In the introduction, we have added information about the RFC and FRα and how they might function together.
Reviewer 2 Report
The systematic review of literature and meta-analysis performed by Rossignol and Frye have comprehensively described the effects of Leucovorin in relation to folate deficiency in ASD patients' brains. The study is well-designed with all required statistical analyses. Therefore, the manuscript can be accepted after fixing certain issues with journal name abbreviations in the Reference section.
Author Response
Reviewer #2
The systematic review of literature and meta-analysis performed by Rossignol and Frye have comprehensively described the effects of Leucovorin in relation to folate deficiency in ASD patients' brains. The study is well-designed with all required statistical analyses. Therefore, the manuscript can be accepted after fixing certain issues with journal name abbreviations in the Reference section.
We thank the reviewer for their analysis and comments. We have fixed the journal name abbreviations.